# Development and Validation of the Attitudes towards Social Robots Scale

**DOI:** 10.3390/healthcare12030286

**Published:** 2024-01-23

**Authors:** Daniel B. Niewrzol, Thomas Ostermann

**Affiliations:** Department of Psychology and Psychotherapy, Witten/Herdecke University, 58452 Witten, Germany; daniel.niewrzol@uni-wh.de

**Keywords:** social robots, attitudes, survey, questionnaire, validation

## Abstract

The idea of artificially created social robots has a long tradition. Today, attitudes towards robots play a central role in the field of healthcare. Our research aimed to develop a scale to measure attitudes towards robots. The survey consisted of nine questions on attitudes towards robots, sociodemographic questions, the SWOP-K9, measuring self-efficacy, optimism, and pessimism, and the BFI-10, measuring personality dimensions. Structural relations between the items were detected using principal components analysis (PCA) with Varimax rotation. Correlations and Analysis of Variance were used for external validation. In total, 214 participants (56.1% female, mean age: 30.8 ± 14.4 years) completed the survey. The PCA found two main components, “Robot as a helper and assistant” (RoHeA) and “Robot as an equal partner” (RoEqP), with four items each explaining 53.2% and 17.5% of the variance with a Cronbach’s α of 0.915 and 0.768. In the personality traits, “Conscientiousness” correlated weakly with both subscales and “Extraversion” correlated with RoHeA, while none the subscales of the SWOP-K9 significantly correlated with RoEqP or RoHeA. Male participants scored significantly higher than female participants. Our survey yielded a stable and convergent two-factor instrument that exhibited convincing validity and complements other findings in the field. The ASRS can easily be used to describe attitudes towards social robots in human society. Further research, however, should be carried out to investigate the discriminant and convergent validity of the ASRS.

## 1. Introduction

The idea of artificially created servants and companions has a long tradition and can be found in various ancient myths and legends, like the Jewish tale of Golem, the Greek legend of Talos, or the talking automaton of Albertus Magnus dating back almost 3000 years ago [1]. Within the context of industrialization in the 19th and 20th centuries, visions of the coexistence of humans and machines have been realized. In particular, the term “robot” was introduced by Karel Čapek in his drama “Rossum’s Universal Robots” in 1920 [2]. From then on, the term was adapted not only from science fiction but also from the scientific literature. According to [3], the number of articles per year indexed in PubMed including the words “robot” or “robotic” has increased from 1990 to 2005 by a factor of 12.

When taking a closer look at what is meant by the term “robot”, a variety of heterogeneous types and physical forms of robots is present, ranging from more functional automated arms to anthropomorphic, socially assistive robots [4]. In contrast to the humanlike image of robots created in films or novels, such as C-3PO and R2-D2 from Star Wars, Bender from Futurama, or Sonny from i-Robot [5], robots actually implemented in healthcare have less human aspects than the aforementioned types. Here, robots, such as the Da Vinci surgery robot, were first used in the 1990s [6]. Since then, healthcare has been the area in which most research on the use of robots has taken place [7]. This might be explained by the various areas of the healthcare system where robots can be deployed, such as surgery [8] or mental health [9], as well as the different kinds of patients or clients, such as pediatric [10] or geriatric patients [11].

Independent of their morphology and their ability to assist people, the reception of robots in the media is often characterized by a certain fear or even an anxiety of a “technology takeover” [12,13,14]. While robots that perform a specific function, such as cleaning or helping, are generally accepted, the reaction to robots that are more human or have near-human forms and behaviors is more reserved and fearful. In particular, this becomes evident, among other things, in the fear that robots will replace humans in the workplace or set themselves up against humanity. On the other hand, there is increasing evidence for the notion that attitudes towards robots, especially in the field of healthcare, are becoming more positive [15]. This rather positive attitude goes so far as to ask why humans could not marry or have sex with a robot, leading to an image of a robot as a companion and carer [16,17]. An important potential factor influencing attitudes towards robots is familiarity with machines and digital society in general.

Nowadays, digital assistants can be found in many households in Western industrialized nations, and a growing number of objects are connected with other devices and systems over the Internet of Things [18]. Based on this development, researchers are already talking about the fact that after the generation of digital natives, the next generation will be a “generation of robotic natives” [19].

From a psychological perspective, attitudes towards social robots or humanlike machines should be considered from a basic perspective of developmental psychology. One distinction children learn very early across cultures is that between animate and inanimate objects. According to [20], children as young as 10 months old begin to distinguish between these two categories. The importance of this distinction can be illustrated, for example, by the fact that animated objects are processed in priority [21,22] and that they are remembered better compared to inanimate objects [23,24,25,26]. This clear distinction, however, is seriously challenged with humanlike machines.

It has long been considered desirable, likely for this reason, to develop machines closely resembling humans, thus aiming to enhance human–machine interaction [27] with the hope of improving attitudes towards machines and establishing a bond with them. According to a widely held view, in doing so, care must be taken not to fall into the “uncanny valley”. This term was coined in [28], where it was assumed that the attitude towards a robot improves with increasing similarity to a human, until a point where it drops abruptly (i.e., the uncanny valley). As the appearance of a robot becomes more and more like that of a human being, acceptance also continues to increase until the point where robots and humans become indistinguishable from each other and the maximum level is reached. Although the hypothesis has found wide appeal in engineering sciences and psychology, it is disputed whether the uncanny valley is a universal phenomenon or only occurs on a case-by-case basis (for a review, see [29]).

With the growing appearance of robots in human society, there is an increased need to assess attitudes towards robots. In line with the growing body of research on human–robot interaction in the last decades [30], several scales have already been developed to explore this field, some of which are specific while others are generic in nature (see [31] for a review). Examples of scales include an item collection on Human–Robot Collaboration used for an acceptance model in an online survey with items, e.g., on perceived usefulness, social implication, and robot anxiety [32]. Other approaches have been more specific, like the 12-item Attitudes toward Cooperative Industrial Robots Questionnaire (ACIR-Q) [33] or the Educational robot attitude scale (ERAS), which included 17 items in 4 subscales [34].

For a longer time, the only available scale measuring general attitudes towards (social) robots was the “Negative Attitudes towards Robots Scale (NARS)” [31], which has been used in a variety of surveys [35,36,37]. Recently, a more balanced scale, “General Attitudes Towards Robots Scale” (GAToRS), was presented [38], measuring positive and negative attitudes towards robots from individual and societal perspectives. Another scale, the 25-item “Attitudes towards Social Robots” (ASOR) questionnaire, also has a more balanced view by introducing the three subscales “ascription of mental capacities”, “ascription of socio-practical capacities”, and “ascription of socio-moral status” [39]. Others, like the Attitudes Towards Human–Robot Intimate Relationship Questionnaire (INT-RO 1), have taken up Levy’s idea of viewing robots as intimate companions and carers/helpers of humans, with 22 items and 4 subscales covering the aspects “Trust”, “Sexuality”, “Intimacy”, and “Acceptance” [40]. However, these constructs are still not analyzed with respect to the underlying factorial structure.

Our research aimed to develop a short but more positive and person-centered approach to measuring attitudes towards robots in everyday life, with a particular emphasis on human–robot interaction, through the “Attitudes towards Social Robots Scale (ASRS)”.

## 2. Materials and Methods

### 2.1. Instruments

#### 2.1.1. Attitudes towards Social Robots Scale (ASRS)

In order to capture positive aspects of attitudes towards social robots, items were generated during a psychological assessment class on test construction with 38 students of psychology at the University of Witten/Herdecke. Their task was to generate and critically discuss items, which then were handed over to an in-house expert panel consisting of two healthcare experts, two psychologists, and one computer scientist. 

Generated items should be related to daily life and include interaction with social robots. In particular, against the background of the theoretical discussion in this scientific field, the areas of care and assistance [41,42], living together and partnership [43,44], and consciousness should be represented in the items [45]. 

Items fulfilling these requirements were selected and compared to existing scales, like the NARS, to which they should be dissimilar. Finally, the following items (English translation) with an explicit focus on the relationship between humans and social robots emerged:I would appreciate an intelligent robot as a friend and helper at my side (ASRS 01);A nurse–robot that takes care of me and looks after me in my old age would be fine (ASRS 02);A human-like robot would be allowed to look after my parents when they get old (ASRS 03);I would let a robot take care of my children (ASRS 04);I believe robots could one day develop a consciousness comparable to humans (ASRS 05);If robots develop a consciousness, then there should be rights for them comparable to human rights (ASRS 06);If there were robots that were indistinguishable from humans, I would consider marrying a robot (ASRS 07);I could also imagine sexual contact with robots (ASRS 08).

A ninth item, “Even if robots develop a consciousness they remain machines and thus second-class ‘living beings’”, was surveyed but finally excluded from the final analysis as it was not considered specific for our purpose (see also the results of the factor analysis).

For all items, a 6-point Likert scale was used, ranging from 1 = absolutely disagree to 6 = absolutely agree. The original German items are provided as Appendix A. 

#### 2.1.2. Sociodemographic Questions

Sociodemographic data, such as gender, age, relationship status, school education, higher education, occupation or employment, monthly net household income, living environment, media use, and origin, were collected using a sociodemographic questionnaire.

#### 2.1.3. SWOP-K9 Questionnaire Measuring Self-Efficacy, Optimism, and Pessimism

According to the findings in [46], optimism and self-efficacy are positively associated with an “affinity for technology”. Thus, we added the SWOP-K9 as an additional questionnaire. Developed by Scholler, Fliege, and Klapp in 1999 [47], it is a modified and abridged German version of the questionnaires for the assessment of self-efficacy [48] and optimism by Scheier and Carver [49]. It consists of 9 items measuring self-efficacy (5 items), optimism, and pessimism (2 items each). The answer categories are given on a 4-point Likert scale from 1 = “does not agree” to 4 = “exactly agree”. For each of the three constructs, the mean value of the scored items is calculated, where high values denote a high level in the respective construct.

Based on a German sample ranging from 17 to 83 years (mean: 45.3 years), the reliability of the scales ranges from Cronbach’s α = 0.53 to 0.91. The SWOP-K9 positively correlates with self-esteem and negatively correlates with general helplessness, shyness, and fearful performance.

#### 2.1.4. Big Five Inventory-10 (BFI-10)

According to [50], personality traits play an important role in human–technology interaction. We thus used the BFI-10 [51] as a short scale to measure the broad personality dimensions of Openness, Conscientiousness, Extraversion, Agreeableness, and Neuroticism, with two items each. Psychometric analyses indicated that the BFI-10 scores have a sufficiently high test–retest reliability of r = 0.75 and are good indicators of the (latent) Big Five dimensions [51,52].

### 2.2. Data Collection and Inclusion and Exclusion Criteria 

From June to September 2018, participants aged at least 18 years were recruited for the survey through direct contact as well as through distribution on social media groups and posted flyers at public places. For data collection, the online survey tool SoSciSurvey was used. For participants who were not able to use the online survey, a classic paper–pencil questionnaire version, which corresponded to the design of the online survey, was used. 

### 2.3. Ethical Considerations

Prior to the start of the survey, all participants were informed about the topic, the voluntary nature of their participation, the confidential handling and pseudonymization of data, and the expected duration of the survey, and they had to confirm that they were providing informed consent. Ethical approval was obtained from the Ethical Committee of Witten/Herdecke University (ID: S-318/2023; approved on 19 December 2023).

### 2.4. Statistical Analysis

Structural relations between the items were detected using principal components analysis (PCA). Prior to this, sampling adequacy and multicollinearity were tested using the Kaiser–Meyer-Olkin criterion (KMO) and Bartlett test of sphericity to ensure appropriateness of the items for principle component analysis. A KMO ≥ 0.50 and a significant Bartlett test of sphericity (*p* < 0.05) were regarded as sufficient. Varimax rotation then was applied to arrive at a solution that satisfies the invariance condition [53] and demonstrates the best and most coherent structure with respect to the amount of explained variance. 

The Kaiser–Gutman criterion (Eigenvalue > 1) and parallel analysis suggested by Horn [54] were used as methods to determine the number of factors. For Horn’s parallel analysis, 1000 parallel datasets were created and compared using their means and the upper 95th percentile, as described in [55].

Internal consistency of the item pool was examined by calculating Cronbach’s alpha and McDonalds Omega coefficients for the factors generated through PCA. The internal reliability was assessed by means of item–total correlations. Item communalities were calculated. Commonly, a communality value of 0.4 is chosen as the cutoff value above which items are retained in a factor analysis [56]. To ensure that outliers did not influence the results, outlier analysis was performed using the Identify Unusual Cases module of SPSS [57]. Cases with an Anomaly index >2 were defined as outliers, and PCA was run again after excluding the outliers to ensure the stability of the generated factors. 

Further statistical analyses included descriptive statistics and the calculation of subscale means in accordance with common recommendations [58]. Additionally, the interrelationship between the ASRS subscales with the SWOP-K9 and the BFI-10 and with sociodemographic parameters like the gender, age, and origin of the participants was calculated using Pearson correlation coefficients, one-factorial Analysis of Variance (ANOVA), and, in cases of variance inhomogeneity, the Kruskal–Wallis H-Test. 

Therefore, the following groups were built: age < 30 years, ≥30 years; relationship status: “In a relationship”, “Single”; higher education: no higher degree, higher degree; monthly income: <EUR 1000, EUR 1000–3000, >EUR 3000; context of socialization (inhabitants): <20,000 (small town), 20,000–100,000 (middle town), >100,000 (city); job area: health sector, social and educational sector, others. A *p*-value of 0.05 was considered to be significant. All statistical analyses were performed using IBM SPSS 29 for Windows (IBM Corp. in Armonk, NY, USA).

### 2.5. Sample Size

According to Osborne and Costello (2009) [59], the required sample size of a factor analytical approach should at least have a subject to item ratio of 10:1 and preferable a ratio of 20:1 to avoid an unstable factor structure. Thus, for 8 items of the ASRS, a minimum of 80 participants is needed, and more than 160 participants is optimal. Thus, due to an estimate of 20% incomplete data, we aimed to reach a sample of at least 200 to arrive at a stable internal consistency. This sample size also enables the detection of moderate between-group effects that are statically significant with a power of 80% and a given level of significance of α = 0.05. 

## 3. Results

### 3.1. Sample

A total of 330 people participated in the survey in the given time frame. In total, 116 of them were excluded due to incomplete datasets or an age below 18 years, which led to *N* = 214 complete records that were included in the evaluation.

In total, 120 (56.1%) of the participants were female and 94 (43.9%) were male. The average age of the participants was 30.8 ± 14.4 years, with a median of 24 and a mode of 22 years. The youngest person was 18 and the oldest 92 years old, so the sample covered a range of 75 years. Furthermore, 79.0% of respondents were between 20 and 60 years old. Most of the participants were either in a relationship (*n* = 81; 37.9%) or married (*n* = 48; 22.4%), while 81 participants (37.9%) stated that they were single. A great majority of the participants (*n* = 173; 80.8%) had a high school diploma. Of those, 68 (31.8%) had a BSc or higher university degree. Almost one third of the sample (*n* = 75; 35.0%) had a monthly income below EUR 1000, while the second third of the sample had a monthly income between EUR 1000 and EUR 3000 (*n* = 73; 34.2%). Moreover, 22.4% (*n* = 48) stated that they earned more than EUR 3000, and 18 participants (8.4%) gave no answer. Most of the participants (*n* = 83; 38.8%) stated that they worked in the health sector, followed by the social sector (*n* = 23; 10.7%), the educational sector (*n* = 21; 9.8%), and public services/authorities (*n* = 17; 7.9%). Other fields were sales and finance (*n* = 12; 5.53%), IT, technology and media (*n* = 7; 3.3%), and R&D (*n* = 7; 3.3%), or the field was not further specified (*n* = 44; 20.6%). In total, 84 participants (39.2%) grew up in a small town with a population below 20,000 inhabitants, *n* = 62 (29.0%) spent their childhood in the context of a middle town (20,000 to 100,000 inhabitants), and 68 participants (31.8%) were raised in a city with more than 100,000 inhabitants. A detailed description of the sample is given in Table 1.

### 3.2. PCA, Reliability, and Validity

Adequacy for PCA was confirmed with a KMO of 0.833 and a significant Bartlett’s test of sphericity (χ^2^(28) = 1036.6, *p* < 0.001). Item–item correlations ranged from 0.285 to 0.901. After three iterations of Varimax rotation, the PCA found two main components with Eigenvalues > 1 and four items, each explaining 70.7% of the variance. Communality values for all eight items were located between 0.496 and 0.900 and thus can be regarded as generally sufficient. 

Horn’s parallel analysis also confirmed this solution, as can be seen in Figure 1, where the Eigenvalue curve of the correlation matrix (scree plot) is compared with that between normally distributed random variables and the upper 95th percentile. 

The first factor explained 53.2% of the variance and included items dealing with care and support delivered by the robot, i.e., “I would appreciate an intelligent robot as a friend and helper at my side” or “A robot that takes care of me and looks after me in my old age would be fine”. Factor loadings of the items ranged between 0.785 and 0.929, with only one side loading >0.3 for the second factor for the item ASRS04 “I would let a robot take care of my children”. With Cronbach’s α = 0.915 and a McDonalds Omega of 0.925, the internal consistency of this factor can be considered excellent. Finally, correlation of the items with the factor ranged between 0.727 and 0.895. The scale was named “Robot as a helper and assistant” (RoHeA).

The second component included items dealing with the robot as an equal partner, i.e., “If robots develop a consciousness, then there should be rights for such machines in addition to human rights” or “If there were robots that were indistinguishable from humans, I would consider marrying a robot”. It explained 17.5% of the variance, with factor loading ranging between 0.680 and 0.843 and without side loadings >0.3 for the first factor. With a Cronbach’s α of 0.768 (Cronbach’s α if the item is deleted between 0.674 and 0.748) and a McDonalds Omega of 0.770, the validity of the second factor is still sufficient with regards to its internal consistency. The correlation of the items with the factor ranged between 0.509 and 0.694. The scale was named “Robot as an equal partner” (RoEqP). It should also be noted that all items were right-skewed, which means that there was rather a tendency to answer the question negatively (the expected mean was 3.5 for all items). The complete results of the factor analysis together with descriptive statistics of the items (M and SD) are displayed in Table 2.

An analysis of outliers using the SPSS routine “Identify Unusual Cases” only detected 11 cases with Anomaly index values > 2. Repeating the analyses after excluding these cases, however, did not notably change the results of the PCA or reliability analysis (two factors explaining 45.9 and 21.2% of the variance; Cronbach’s alphas: 0.898 for the first and 0.700 for the second factor).

### 3.3. Relationship with Other Constructs

To detect differences in the two ASRS subscales with respect to sociodemographic data, ANOVA was performed. In cases of variance inhomogeneity, the Kruskal–Wallis H-Test was additionally applied to the data. Table 3 summarizes the results. 

A gender-related difference was found in both scales, with male participants scoring significantly higher than female participants. We also found that younger participants (<30 years) could significantly better imagine having a robot as an equal partner than older participants (*p* = 0.029). This was also true for participants who described themselves as “single” compared to those in a relationship. Moreover, although there were some slight differences with respect to higher education, income, and context of socialization, we were not able to detect significant differences between the respective subgroups. 

Variance inhomogeneity was detected in both scales with respect to gender in the RoEqP scale for the relationship status and in the RoHeA scale for income as an independent variable. Except for the RoEqP scale for the relationship status, which was no longer significant (*p* = 0.064), the results of the ANOVA were confirmed using the Kruskal–Wallis H-Test.

### 3.4. Relationship with Other Constructs

All correlation analyses show no or only low correlations between the two scales, RoEqP and RoHeA, and the other constructs. Only the scale “Conscientiousness” correlated significantly negatively with both RoEqP (r = −0.206; *p* < 0.01) and RoHeA (r = −0.210; *p* < 0.01), while “Extraversion” correlated significantly negatively only with RoHeA (r = −0.186; *p* < 0.01). All other personality traits did not show a correlation to the robot scales. Also, the subscales “Pessimism”, “Optimism”, and “Self-efficacy” of the SWOP-K9 did not significantly correlate with RoEqP and RoHeA (Table 4).

## 4. Discussion

Social robots have become increasingly popular in recent years, and it is of paramount importance to determine what people’s attitudes towards them are. There is a growing body of literature exploring various aspects of interactions between social robots and humans, including everyday situations as well as the impact on education and healthcare [31,60,61]. According to a recent scoping review of forty-three articles, two thirds of the included articles found a rather positive attitude towards social robots in the field of health. However, they also pointed towards the difficulty of measuring “acceptability under ecological conditions and over the long term” [62]. Moreover, according to [63], there is an urgent need for research on scales and questionnaires quantifying attitudes towards social robots.

This article aims at contributing to this field of research and summarizes the first results of the development and validation of the Attitudes towards Social Robots Scale (ASRS). Our factor analytic tests yielded a stable and convergent two-factor solution that exhibited convincing validity, with values of Cronbach’s alpha of 0.915 and 0.768.

Our results regarding the correlation of personality traits and attitudes towards robots somehow deviate from findings in [64], who found Agreeableness, Extraversion, and Openness but not Conscientiousness to be related to attitudes towards robots. They are also different from the findings in [40], where, in particular, the INT-RO-1 subscale “Trust” statistically significantly predicted Agreeableness. 

On the other hand, another study found that that Neuroticism correlated with the NARS subscales “Negative attitude towards situations of interaction with robots” (r = 0.259, *p* < 0.001), “Negative attitude towards the social influence of robots” (r = 0.137, *p* = 0.039), and “Negative attitude towards emotions in interaction with robots” (r = 0.142, *p* = 0.031) [65]. They also found a positive correlation between Extraversion and the subscale “Negative Attitudes towards social influence of robots” (r = 0.146, *p* = 0.027), which could be replicated in our study.

Findings in [66], however, are only partly in line with our results, as they also found no relationship with “Neuroticism” (r = 0.04), but, in contrast to our findings, they found a significant association of Conscientiousness with “Negative attitudes towards robots”. However, it remained unclear which subscale of the NARS was used.

With respect to our scales, namely having a robot as a companion, a survey of visitors to a technology exhibition in Switzerland found that people had a positive attitude towards the idea of sharing their daily live with robots [67]. Female respondents in this survey were “much less willing to accept robot technologies in their life” than male visitors, which is congruent with our study results. Results also indicated that elderly people were more skeptical towards owning a robot that would do their daily tasks. On the other hand, a robot helping to maintain their autonomy in daily life was rated positively.

A more differentiated picture on the acceptance of service robots by elderly people and nursing staff was found in [68]. Again, men were more positive than women, but participants aged over 65 years did not differ from those between 40 and 65 years with respect to their attitudes towards healthcare robots, which is also confirmed in our results.

A German survey additionally demonstrated that half of the seniors and nursing staff reported positive attitudes regarding service robots for the elderly [69]. However, about 40% of participants in that study were clearly against service robots in their daily life, potentially because they did not feel safe handling a robot, particularly with respect to potential failures of the robot.

Another survey investigated perceptions and emotions towards the utilization of a healthcare robot in a questionnaire survey subsequently followed by a measurement of participants’ blood pressure assessed by a robot or a human assistant [70]. Although no difference in blood pressure level was obtained between the two groups, participants stated that they were more comfortable when treated by a medical student.

In the multilevel modelling approach using data from the Eurobarometer, the effects of age and level of education were not significant with respect to attitudes towards social robots [50]. However, they found a significant association between individuals living in big cities and belonging to a high social class: they were “more likely to indicate a favorable acceptance of robots for the provision of companionship”, which is not supported by our findings, which might be due to the differences in the populations that were surveyed. 

Some of the above findings are also summarized in a recent meta-analysis of 26 publications [71], which points out that there are various, partly conflicting findings regarding the relationship between attitudes towards robots and other factors, such as psychological differences in personality, or sociodemographic variables, like gender, age, prior education or robot experience, and global region. In particular, they found that the more agreeable, extroverted, and open individuals are, the more likely they are to accept a robot. 

In addition, no evidence was found to support an association between Neuroticism or Conscientiousness and the willingness to accept a robot. Interestingly, human personality traits were only significant predictors for the acceptance of robots in younger samples from 18–24 years. Finally, robot acceptance and human personality only correlated significantly in European studies. Thus, studies should be performed in mixed and multinational populations to obtain a clearer picture of factors that are likely to influence or mediate relationships with robots.

## 5. Limitations

Our study has a number of limitations. Firstly, the item selection did not rely on qualitative studies in advance. Other studies prepared the scale development through preliminary studies or intensive conceptual theory models, which was not performed in this study. This, however, does not imply that the scale is not valid, which is also supported by psychometric results, which show convincing values. 

Secondly, and unlike in [35], no differentiation between different forms and types of robots and their specifications was made. However, from the item formulation, it could be easily derived that our approach was focused on humanoid robots.

As another limitation, the NARS measures aspects of fear that may prevent people from interacting with robots by including items like “I would feel uneasy if robots really had emotions” or “I would feel paranoid talking with a robot”. Our approach was conceptualized to be more positive. Therefore, it is very likely that our scale only captures emotions, such as fear or restraint, to a very limited extent. An external validation comparing the ASRS to NARS and the GAToRS could therefore reveal new insights into the nature our findings and should be carried out in future surveys.

Moreover, the data presented here are from 2018, and attitudes towards robots may have changed and influenced the scale structure found here. However, research on measurement invariance over time of scales and constructs, e.g., in the field of middle school students’ attitudes towards computer programming [72], which might show a similar development, reveals that although scale mean values might change, the structure below is stable. 

From a methodological point of view, the use of an exploratory factor analysis approach (EFA) can be criticized in favor of a Confirmatory factor analysis approach (CFA), in particular in the construction of psychological constructs [73]. While EFA is a hypothesis-generating method to explore a possible underlying structure and explain the same amount of variance with fewer variables, CFA is a hypothesis-testing method that aims at verifying a given factor structure of a set of observed variables [74]. 

For scale development studies, it thus is recommended that an exploratory approach should be used prior to CFA to test the validity of the structure [56,75], as researchers might be incorrect in their assignment of items to the factors, which can lead to supposedly convincing but unreliable constructs [76]. 

The question of whether an exploratory approach should be used is still open [77]. While in [78] it was shown that a PCA solution compared to a principal axis factoring approach (PAF) had more items with cross loadings, a simulation study found that PCA loadings might be better approximations of the true factor loadings than the loadings produced by PAF [79]. Thus, we believe that our approach of using a PCA in the first instance is sufficiently covered by the existing literature. Finally, and in accordance with [80], we decided against a PCA subsequently followed by a CFA using the same data, as it might produce overfitting.

In this context, the adequate sample size might also be a matter of discussion [81]. Although no simple approaches to sample size determination exist for PCA, it should be noted that the number of cases in the present study is in the order of magnitude of comparable studies. Moreover, according to [82], the half width of the 95% confidence interval of Cronbach’s alpha coefficient for a 10-item instrument at a sample size of 200 is approximately 0.02 for a Cronbach’s alpha of 0.9 and 0.04 for a Cronbach’s alpha of 0.8. Thus, taking these approximations into account, the internal validity of our scale is sufficiently guaranteed.

## 6. Conclusions

In sum, the ASRS is a sound survey instrument to economically assess attitude towards robots. It can be used efficiently in different areas of public life and healthcare. Due to its compact format, it is easy and quick to answer, and it can therefore be combined with other instruments without any problems. However, more studies are needed using different samples, such as nurses or elderly people for whom such technology would be directly relevant. Based on these studies, it would be necessary to test whether the scale shows enough measurement invariance. 

## Figures and Tables

**Figure 1 healthcare-12-00286-f001:**
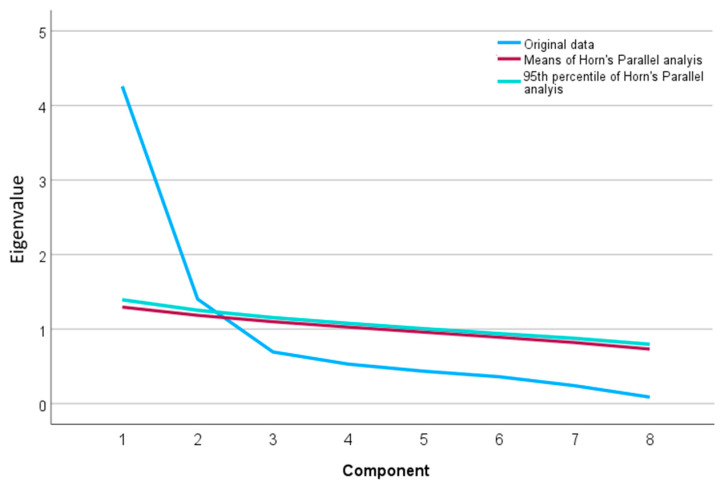
Scree plot of the Eigenvalues of the original data (blue line) compared to the mean Eigenvalues from 1000 normally distributed random datasets.

**Table 1 healthcare-12-00286-t001:** Sociodemographic data subdivided by gender.

	Male (*n* = 94; 43.9%)	Female (*n* = 120; 56.1%)	Total (*N* = 214)
Age			
Mean ± SD	31.2 ± 13.8	30.6 ± 14.8	30.8 ± 14.4
Median	24.5	23.0	24.0
Min	18	18	18
Max	92	91	92
Relationship status			
In a relationship	29 (30.9%)	52 (43.3%)	81 (37.9%)
Single	44 (46.8%)	37 (30.8%)	81 (37.9%)
Married	20 (21.3%)	28 (23.3%)	48 (22.4%)
Divorced	1 (1.1%)	3 (2.5%)	4 (1.9%)
Widowed	0 (0.0%)	1 (0.8%)	1 (0.5%)
Other	0 (0.0%)	1 (0.8%)	1 (0.5%)
Education (School)			
Secondary school	5 (5.3%)	5 (4.2%)	10 (4.7%)
Secondary modern school	12 (12.8%)	12 (10.0%)	24 (11.2%)
High school	74 (78.2%)	99 (82.5%)	173 (80.8%)
Other schools	0 (0.0%)	1 (0.8%)	1 (0.5%)
No answer	3 (3.2%)	3 (2.5%)	6 (2.8%)
Higher Education			
No higher degree	54 (57.4%)	77 (64.2%)	131 (61.2%)
Bachelor’s degree	9 (9.6%)	11 (9.2%)	20 (9.3%)
Master’s degree or similar	12 (12.8%)	22 (18.3%)	34 (15.9%)
PhD or similar	7 (7.4%)	0 (0.0%)	7 (3.3%)
Other	5 (5.3%)	2 (1.7%)	7 (3.3%)
No answer	7 (7.4%)	8 (6.7%)	15 (7.0%)
Monthly income			
<EUR 1000	23 (24.5%)	52 (43.3%)	75 (35.0%)
EUR 1000–2000	21 (22.3%)	23 (19.2%)	44 (20.6%)
EUR 2000–3000	14 (14.9%)	15 (12.5%)	29 (13.6%)
EUR 3000–5000	17 (18.1%)	14 (11.7%)	31 (14.5%)
>EUR 5000	10 (10.6%)	7 (5.8%)	17 (7.9%)
No answer	9 (9.6%)	9 (7.5%)	18 (8.4%)
Raised in (inhabitants)			
<20,000	42 (44.7%)	42 (35.0%)	84 (39.2%)
20,000–100,000	27 (28.7%)	35 (29.2%)	62 (29.0%)
>100,000	25 (26.6%)	43 (35.8%)	68 (31.8%)
Job area			
Health Sector	40 (42.6%)	43 (35.8%)	83 (38.8%)
Social Sector	4 (4.3%)	19 (15.8%)	23 (10.7%)
Education	8 (8.5%)	13 (10.8%)	21 (9.8%)
Public Services/Authorities	7 (7.4%)	10 (8.3%)	17 (7.9%)
IT/Tech Sector/Media	6 (6.4%)	1 (0.8%)	7 (3.3%)
Sales and Finance	3 (3.2%)	9 (7.5%)	12 (5.6%)
R&D	2 (2.1%)	5 (4.2%)	7 (3.3%)
Others	24 (25.5%)	20 (16.7%)	44 (20.6%)

**Table 2 healthcare-12-00286-t002:** Results of the PCA, reliability, and items parameters (*M* = Mean; *SD* = standard deviation). Scale range from 1 = completely disagree to 6 = completely agree). Factor loadings > 0.6 are marked in bold.

Item Codes	Components		Corrected Item–Scale Correlation	Items’ Reliability		Item Parameters	
	1	2		Cronbach’s Alpha, If Item Is Deleted	Communalities	*M*	*SD*
ASRS 01	**0.826**	0.170	0.726	0.917	0.711	2.33	1.50
ASRS 02	**0.929**	0.190	0.895	0.843	0.900	2.47	1.59
ASRS 03	**0.905**	0.262	0.888	0.842	0.888	2.27	1.54
ASRS 04	**0.785**	0.308	0.733	0.915	0.711	1.76	1.25
ASRS 05	0.183	**0.680**	0.509	0.748	0.496	2.92	1.57
ASRS 06	0.189	**0.758**	0.592	0.710	0.611	2.55	1.74
ASRS 07	0.185	**0.843**	0.694	0.674	0.744	1.56	1.12
ASRS 08	0.236	**0.736**	0.554	0.722	0.597	1.60	1.32
Cronbach’s Alpha	0.915	0.768					
McDonalds Omega	0.925	0.770					

**Table 3 healthcare-12-00286-t003:** Differences in the two ASRS subscales with respect to sociodemographic data (in cases of variance inhomogeneity, the Kruskal–Wallis H-Test was additionally applied to the data).

	RoHeA	RoEqP
Gender		
Male	2.61 ± 1.48	2.44 ± 1.31
Female	1.88 ± 1.07	1.93 ± 0.88
F, *p*-Value	F = 17.329; *p* < 0.001	F = 11.514; *p* < 0.001
H, *p*-Value	H = 12.939; *p* < 0.001	H = 6.141; *p* = 0.013
Age		
<30 years	2.17 ± 1.27	2.27 ± 1.17
≥30 years	2.28 ± 1.41	1.92 ± 0.98
F, *p*-Value	F = 0.602; *p* = 0.557	F = 5.95; *p* = 0.029
Relationship status		
“In a relationship”	2.12 ± 1.25	2.00 ± 0.94
“Single”	2.33 ± 1.40	2.38 ± 1.31
F, *p*-Value	F = 1.35; *p* = 0.246	F = 6.32; *p* = 0.013
H, *p*-Value		H = 3.426; *p* = 0.064
Higher education		
No higher degree	2.14 ± 1.18	2.10 ± 1.06
Higher degree	2.30 ± 1.51	2.31 ± 1.22
F, *p*-Value	F = 0.70; *p* = 0.405	F = 1.66; *p* = 0.199
Monthly income		
<EUR 1000	2.31 ± 1.42	2.23 ± 1.14
EUR 1000–2000	1.95 ± 1.11	2.22 ± 1.17
EUR 2000–3000	1.88 ± 1.15	1.77 ± 0.91
EUR 3000–5000	2.14 ± 1.14	2.20 ± 1.04
>EUR 5000	2.90 ± 1.75	2.24 ± 1.19
No answer	2.40 ± 1.20	2.19 ± 1.30
F, *p*-Value	F = 1.86; *p* = 0.103	F = 0.82; *p* = 0.539
χ^2^, *p*-Value	H = 7.524; *p* = 0.184	
Context of socialization (inhabitants)		
<20,000 (Small town)	2.34 ± 1.41	2.15 ± 1.25
20,000–100,000 (Middle town)	2.14 ± 1.20	2.06 ± 1.00
>100,000 (City)	2.11 ± 1.30	2.25 ± 1.06
F, *p*-Value	F = 0.68; *p* = 0.510	F = 0.45; *p* = 0.642
Job area		
Health sector	2.19 ± 1.29	2.10 ± 1.11
Social and educational sector	1.91 ± 1.12	1.95 ± 1.02
Others	2.37 ± 1.42	2.32 ± 1.17
F, *p*-Value	F = 1.81; *p* = 0.167	F = 1.76; *p* = 0.174

**Table 4 healthcare-12-00286-t004:** Pearson correlation coefficients (r) between the two ASRS subscales and the BFI-10 and the SWOP-K9 subscales with 95% confidence intervals. Significant correlations are marked in bold.

	ASRS Subscales	
	RoHeA r [95%CI]	RoEqP r [95%CI]
BFI-E (Extraversion)	**−0.186 [−0.312; −0.053]**	−0.043 [−0.177; 0.091]
BFI-N (Neuroticism)	0.057 [−0.078; 0.189]	0.027 [−0.108; 0.160]
BFI-O (Openness)	−0.071 [−0.203; 0.064]	0.042 [−0.092; 0.175]
BFI-G (Conscientiousness)	**−0.210** [−0.335; −0.078]	**−0.206 [−0.331; −0.074]**
BFI_V Agreeableness	0.073 [−0.061; 0.205]	0.007 [−0.127; 0.141]
SWOP-K9 Optimism	−0.046 [−0.179; 0.089]	−0.018 [−0.152; 0.116]
SWOP-K9 Pessimism	0.051 [−0.083; 0.184]	0.040 [−0.095; 0.173]
SWOP-K9 Self-efficacy	0.049 [−0.086; 0.182]	0.073 [−0.062; 0.205]

## Data Availability

The data presented in this study are available upon request from the corresponding author.

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
