# Peer review of "Development and Validation of the Attitudes towards Social Robots Scale"

_healthcare, 2024, doi:10.3390/healthcare12030286_

Round 1

Reviewer 1 Report (Previous Reviewer 3)

Comments and Suggestions for Authors

The revisions look fine. There is some scope for further proofreading e.g. Abstract: "Conscientiousness missing closing quotation marks; Table 2: some decimal values indicated with comma instead of decimal point; Common Factor Analysis > Confirmatory Factor Analysis - but all points addressed.

Author Response

Authors comment: Thank you once again for having a closer look at the formal aspects. We went through the manuscript again and tried to eliminate the inaccuracies. With respect to the term CFA we meant “Common Factor Analysis”. To avoid misunderstandings we rephrased to “Exploratory Factor Analysis” (EFA) but however added another chapter on “Confirmatory Factor Analysis”(CFA).

Reviewer 2 Report (New Reviewer)

Comments and Suggestions for Authors

I appreciate the opportunity to review this article. The topic is interesting, but some elements still raise some concerns:

  1. It would be useful to indicate the number of experts from each area who analyzed the items proposed by the students. Additionally, the authors should highlight the experts’ contributions to the items created. Furthermore, I recommend that the authors consult the relevant literature to establish a connection between the content of each item in the questionnaire and the theme of “attitudes toward Sociable Robots.”

  1. The instrument scores were calculated based on the average of the responses. However, since the response alternatives come from a Likert scale, it is inappropriate to use the average of these responses to generate scores.

  1. While it was considered that only 8 items were applied to ensure a good sample size, it appears that other instruments were also used in parallel. This information leaves the article somewhat confusing regarding sample size estimation.

  1. The authors stated that the KMO test verifies multicollinearity. However, this information appears to be inaccurate.

  1. It is unclear why the commonality of the items was not analyzed.

  1. Additionally, the rationale behind using two dimensions (rather than one or three) is not well-explained. The authors could present a parallel analysis graph (or screeplot) to illustrate the eigenvalues related to each probable dimension of the instrument.

  1. It remains unclear why sociodemographic data was analyzed by sex. Additionally, there is no comparison regarding the heterogeneity of responses given by the two biological sexes.

  1. In general, the correlation values presented in Table 3 are low. Consequently, there appears to be no correlation between the scores of the different instruments.

  1. The article does not address whether the data is free of outliers. Furthermore, it does not guarantee the absence of multicollinearity in the items. The convergent and discriminant validity at both the instrument level and item level remains unknown.

  1. While the items exhibit good factor loading values, determining the level of attitudes of an individual toward Sociable Robots remains a challenge. When applying this instrument to a person, how do we ascertain their specific level of attitudes toward Sociable Robots?

  1. It is essential to ensure statistical assumptions when using the ANOVA test. Unfortunately, the article does not provide information on these statistical assumptions.

  1. The discussion does not clearly articulate the gaps filled by the new instrument (and its scales) in comparison to other instruments present in the literature.

  1. The practical and theoretical implications of this study are not easily identifiable.

  1. In the limitations section, one paragraph compares AFC and PCA. However, these procedures serve different purposes. CFA (Confirmatory Factor Analysis) is used to measure reflective latent traits, while PCA (Principal Component Analysis) is appropriate for formative variables. The authors may not be aware of whether their scales are formative or reflective.

  1. Another limitation would be to validate the instrument based on the Item Response Theory (reference Barbosa et al.,2021). Likewise, the framework could be validated and applied alongside a theoretical model through structural equation modeling based on variance (reference Souza et al. 2021) or based on covariance (reference Bianco et al., 2023).

Bianco, D., Bueno, A., Godinho Filho, M., Latan, H., Ganga, G. M. D., Frank, A. G., & Jabbour, C. J. C. (2023). The role of Industry 4.0 in developing resilience for manufacturing companies during COVID-19. International Journal of Production Economics, 256, 108728.

Souza Barbosa, A. da, da Silva, L. B., Morioka, S. N., da Silva, J. M. N., & de Souza, V. F. (2021). Item response theory-based validation of an integrated management system measurement instrument. Journal of cleaner production, 328, 129546.

Souza, D. S. F. de, da Silva, J. M. N., de Oliveira Santos, J. V., Alcantara, M. S. B., & Torres, M. G. L. (2021). Influence of risk factors associated with musculoskeletal disorders on an inner population of northeastern Brazil. International Journal of Industrial Ergonomics, 86, 103198.

Author Response

It would be useful to indicate the number of experts from each area who analyzed the items proposed by the students. Additionally, the authors should highlight the experts’ contributions to the items created. Furthermore, I recommend that the authors consult the relevant literature to establish a connection between the content of each item in the questionnaire and the theme of “attitudes toward Sociable Robots.”

Authors comment: We added the numbers of experts from each area who analyzed the items. As the format was a discussion round, we however are unable to detect the experts’ contributions to the items created. We also added further literature to establish a connection between the content of the items in the questionnaire and the theme of “attitudes toward Sociable Robots in the methods section.”

The instrument scores were calculated based on the average of the responses. However, since the response alternatives come from a Likert scale, it is inappropriate to use the average of these responses to generate scores.

Authors comment: In almost all cases questionnaire scores are either represented as sum scores or means of Likert scaled items. Thus, we are a bit surprised about this comment and would like to stay in line with our approach which is also underpinned by well cited literature (e.g. Norman, G. (2010). Likert scales, levels of measurement and the “laws” of statistics. Advances in health sciences education, 15, 625-632). 

While it was considered that only 8 items were applied to ensure a good sample size, it appears that other instruments were also used in parallel. This information leaves the article somewhat confusing regarding sample size estimation.

Authors comment: The other items were not an issue for factor analysis and thus are not relevant for sample size estimation. The use of other instruments for inferential purposes is a common procedure and as described has sufficient statistical power with the given sample size.

The authors stated that the KMO test verifies multicollinearity. However, this information appears to be inaccurate.

Authors comment: In our methods section we wrote: “…sampling adequacy and multicollinearity were tested using the Kaiser-Meyer-Olkin criterion: (KMO) and Bartlett test of sphericity to ensure appropriateness of the items for principle component analysis. A KMO ≥ 0.50 and a significant Bartlett test of sphericity (p < 0.05) were regarded as sufficient.” In the results we stated: “Adequacy for PCA was confirmed with a KMO of .833 and a significant Bartlett’s test of sphericity (χ2(28) = 1036.6 p < .001).” We however did not state that “the KMO test verifies multicollinearity”.

It is unclear why the commonality of the items was not analyzed.

Authors comment: Thank you for this helpful comment. We now provide the respective information in Table 2 and discuss the commonality of the items in the discussion part.

Additionally, the rationale behind using two dimensions (rather than one or three) is not well-explained. The authors could present a parallel analysis graph (or screeplot) to illustrate the eigenvalues related to each probable dimension of the instrument.

Authors comment: Thank you again for this very helpful comment. We originally used the Kaser Guttman Criterion (Eigenvalues >1) but now added a parallel analysis and added the respective chapters in the methods and discussion part in accordance with your suggestion.

It remains unclear why sociodemographic data was analyzed by sex. Additionally, there is no comparison regarding the heterogeneity of responses given by the two biological sexes.

Authors comment: Thank you for this helpful comment. Indeed, a  gender related analysis was not the major topic of this paper and we would like to abstain from a detailed analysis of gender related issues. Table one now offers other researchers to opportunity to use gender specific results, i.e. in reviews or meta analyses. However, we can of course modify table 1 and delete the respective columns if this is a necessary condition.

In general, the correlation values presented in Table 3 are low. Consequently, there appears to be no correlation between the scores of the different instruments.

Authors comment: Thank you for pointing this out. Indeed the correlations are low and as described “were not linked to the robot scales”. We now have rephrased the respective part to be more clear. 

The article does not address whether the data is free of outliers. Furthermore, it does not guarantee the absence of multicollinearity in the items. The convergent and discriminant validity at both the instrument level and item level remains unknown.

Authors comment: Multicollinearity of the items in our view was sufficiently addressed (see 3rd comment). Large outliers according to (Liu et al. 2012) in severe cases may create an extra factor or might reduce the number of factors in EFA. We have addressed this point by analyzing anomalities in the data set and identified 11 cases with high anomality indices >2. This however did not change the results of the factor analysis  when we excluded them from our analysis. We thus would like to leave the analysis in it’s present form. With respect to convergent and discriminant validity at both the instrument level and item level we indeed only have the present analysis. We however have discussed this point in the limitation section and also added this into the abstract.      

While the items exhibit good factor loading values, determining the level of attitudes of an individual toward Sociable Robots remains a challenge. When applying this instrument to a person, how do we ascertain their specific level of attitudes toward Sociable Robots?

Authors comment: Indeed there are no cutoff values or other markers which quantify the personal level of attitudes toward Sociable Robots (which however is also true for other scales in the field of social robots). We thus understand our work as a starting point and have pointed out this aspect in the discussion part.

It is essential to ensure statistical assumptions when using the ANOVA test. Unfortunately, the article does not provide information on these statistical assumptions.

Authors comment: Indeed homogeneity of variances is a crucial point. We only found three violations of this assumption and in such cases used the nonparametric Kruskal –Wallis Test as additional test. We modified the methods section accordingly. We however refer again to (Norman, G. (2010)) who clearly stated that non parametric test have a smaller statistical power and discuss this point.    

The discussion does not clearly articulate the gaps filled by the new instrument (and its scales) in comparison to other instruments present in the literature.

Authors comment: The major new aspect lies in the personal approach with a more neutral and positive connotation of the items. Another aspect is it’s shortness. We have already pointed this out.

The practical and theoretical implications of this study are not easily identifiable.

Authors comment: As this article is on scale development only, the practical and theoretical implications of this study will be visible when applying it in interaction contexts of humans with social robots. We have added some implications in the discussion section  

In the limitations section, one paragraph compares AFC and PCA. However, these procedures serve different purposes. CFA (Confirmatory Factor Analysis) is used to measure reflective latent traits, while PCA (Principal Component Analysis) is appropriate for formative variables. The authors may not be aware of whether their scales are formative or reflective.

Authors comment: Reviewer 1 already pointed on this aspect and there is probably a misinterpretation. We wanted to show that PCA may not be different from other (“common”) factor analytic approaches. Thus, we did not want to discuss Confirmatory Factor Analysis and PCA but Common factor analysis and PCA. We now renamed it into Exploratory Factor analysis to be more precise and added some more references. We however understand, that a discussion of Confirmatory Factor Analysis versus PCA is necessary. Thus we added a respective chapter on this issue. Im particular we argue in accordance with (Fokkema and Greiff, 2017) that a PCA and a CFA on the same data should be avoided and that in cases where nothing is known a PCA is suitable.

Another limitation would be to validate the instrument based on the Item Response Theory (reference Barbosa et al.,2021). Likewise, the framework could be validated and applied alongside a theoretical model through structural equation modeling based on variance (reference Souza et al. 2021) or based on covariance (reference Bianco et al., 2023).

Authors comment: Item response theory (IRT), also known as the latent response theory refers to a family of mathematical models that attempt to explain the relationship between latent traits (unobservable characteristic or attribute) and their manifestations (i.e. observed outcomes, responses or performance). However, we do not have manifestations in terms of attitudes towards social robots. Thus IRT might be applied in further studies when i.e. behavior towards social robots is directly observable.

Reviewer 3 Report (New Reviewer)

Comments and Suggestions for Authors

Dear Authors,

I attached my comments and comments on your document.

Does the introduction provide sufficient background and include all relevant references?

The introduction to the provides a comprehensive background to the development of robots, particularly in health care, and the importance of their attitudes. It incorporates a historical perspective, tracing the concept of artificial servants and companions from ancient myths to industrialization in the 20th century. The term "robots" and their evolution in literature and scientific research are well documented, with references to key works and studies. The introduction effectively sets the context for ASRS development by discussing the heterogeneity of robots, their representation in the media and the psychological aspects of human-robot interaction. It addresses the need for research in this area and examines existing scales of attitudes towards robots, laying a solid foundation for the objectives of the study. The references provided are relevant and appear to cover the necessary literature, supporting the comprehensiveness of the introduction to set the stage for the study.

Are all the cited references relevant to the research :

The document provided is extensive, and it includes a large number of references throughout its sections and they are OK.

 Is the research design appropriate

The research design appears to be appropriate to meet the objectives of the study. It includes a systematic approach to developing and validating a new scale (ASRS) for measuring attitudes towards social robots. This methodology includes a well-structured process, including the generation of items, pilot tests and statistical analyses for validation. The use of a variety of participants and robust statistical methods improves the reliability and validity of the conclusions. This comprehensive approach is well aligned with the objectives of the study and ensures a comprehensive study of attitudes towards social robots in different contexts.

Are the methods adequately described

The method section of the document is detailed and well articulated, providing a clear description of the processes used to develop and validate the social robot scale attitude (ASRS). It contains a complete explanation of the generation of objects, pilot tests and statistical validation techniques. The procedures for collecting and analysing data are also fully described, which ensures transparency and replication of research.

Are the results clearly presented

The method section of the document is detailed and well-defined and provides a clear description of the processes used to develop and validate the Social Robotic Scale Attitude (ASRS). It contains detailed explanations of object generation, pilot tests, and statistical validation technologies. Procedures for collecting and analysing data are also fully described, ensuring the transparency and replication of research.

Are the conclusions supported by the results

The conclusions of the document are well supported by the results. The results of statistical analysis and validation processes provide a solid basis for conclusions about attitudes to the social robots scale (ASRS). The conclusions logically derive from the presented data, demonstrating the reliability and validity of the scale in measuring attitudes to social robots

Major Comments;

Rewrite the abstract

The abstract is not suitable, please review and rewrite the abstract in order to improve. Do not start directly by providing detailed results without explaining what is going on in the article. It is advisable to make improvements to increase clarity and lucidity. The research objective, in particular the development and validation of attitudes towards the social robots scale (ASRS), should be clearly indicated. Including a short overview of the methods used, emphasizing any unique approach, will add value. Key findings, particularly important results concerning ASRS psychometric properties, should be summarized succinctly. In conclusion, with the broader implications of these findings, particularly in the fields of health care and related areas, a comprehensive, but concise overview of the content and significance of the paper will be provided.

Comments on the Quality of English Language

Minor Improvements

In 72- Probably therefore

This sentence could be improved for clarity and grammatical correctness. The phrase "Probably therefore" is awkwardly placed and could be more effectively structured. A revised version might be:

"It has long been considered desirable, likely for this reason, to develop machines closely resembling humans, aiming to enhance

In 47 Issue Fragmented Sentences and Clarity Issue

"Independent of their morphology and their ability to assist people the reception of robots in media is often characterized by a certain fear or even anxiety of a “technology takeover” [12-14]."

Issue: The sentence lacks clarity and is fragmented, leading to confusion about the intended meaning​​

In368 Issue : Inconsistent Hyphenation and Ambiguity

"Firstly the item selection did not rely on qualitative studies in advance."

Issue: The hyphenation in "qualitative" is misplaced and causes unnecessary interruption in reading. The sentence also starts abruptly, which might confuse the reader about its context and do not use I , we our in academic journals unless its necessary.

In 137 Awkward Phrasing and Lack of Clarity

A ninth item “Even if robots develop a consciousness they remain machines and thus second-class 'living beings'” was surveyed

The sentence is lengthy and convoluted, making it hard to follow consider simplify and rewrite it.

Author Response

The abstract is not suitable, please review and rewrite the abstract in order to improve. Do not start directly by providing detailed results without explaining what is going on in the article. It is advisable to make improvements to increase clarity and lucidity. The research objective, in particular the development and validation of attitudes towards the social robots scale (ASRS), should be clearly indicated. Including a short overview of the methods used, emphasizing any unique approach, will add value. Key findings, particularly important results concerning ASRS psychometric properties, should be summarized succinctly. In conclusion, with the broader implications of these findings, particularly in the fields of health care and related areas, a comprehensive, but concise overview of the content and significance of the paper will be provided.

Authors comment Thank you for this comment. We however are a bit puzzled, as the abstract stars with a brief introduction and clearly describes the methods and the results. We however modified the abstract and hop it is now suitable.

In 72- Probably therefore

This sentence could be improved for clarity and grammatical correctness. The phrase "Probably therefore" is awkwardly placed and could be more effectively structured. A revised version might be: "It has long been considered desirable, likely for this reason, to develop machines closely resembling humans, aiming to enhance

            Authors comment: We have changed the respective sentence.

In 47 Issue Fragmented Sentences and Clarity Issue: "Independent of their morphology and their ability to assist people the reception of robots in media is often characterized by a certain fear or even anxiety of a “technology takeover” [12-14] Issue: The sentence lacks clarity and is fragmented, leading to confusion about the intended meaning​​

Authors comment: We have added an “an” before anxiety in the respective sentence which probably was the cause of confusion.

In368 Issue : Inconsistent Hyphenation and Ambiguity: "Firstly the item selection did not rely on qualitative studies in advance." Issue: The hyphenation in "qualitative" is misplaced and causes unnecessary interruption in reading. The sentence also starts abruptly, which might confuse the reader about its context and do not use I , we our in academic journals unless its necessary.

            Authors comment: We changed the wording according to your suggestions.

In 137 Awkward Phrasing and Lack of Clarity: A ninth item “Even if robots develop a consciousness they remain machines and thus second-class 'living beings'” was surveyed”. The sentence is lengthy and convoluted, making it hard to follow consider simplify and rewrite it.

Authors comment: We tried to rephrase but still think this is a clear description. But probably the editorial team of the publisher has a suggestion.

This manuscript is a resubmission of an earlier submission. The following is a list of the peer review reports and author responses from that submission.

Round 1

Reviewer 1 Report

Comments and Suggestions for Authors

The manuscript describes the development of a new scale to measure attitudes toward social robots predominantly in the context of healthcare and the results of a study with 217 participants reporting factor analysis and correlation with other constructs. In principle, of course, the attempt to develop a scale is worthwhile, but the procedure has many major weaknesses in the development and the attempt to validate it. In general, I think that the scale was not properly constructed as it was neither guided by theoretical models nor on the basis of qualitative empirical experience, and also the validation process is based on too little data (only one study with an online convenience sample of only 217 participants seems very small). I am thus not convinced that the scale is reliable or valid. I think, therefore, a lot of work is still necessary before publication including the collection of further data in additional studies. I think that the necessary improvements I propose cannot be made in a reasonable period of time. I thus unfortunately recommend a rejection. In the following I give tips for improvement, which I hope will help the authors in their future scale construction and validation.

1.      1. Title: First, I think the title and the name of the scale are misleading. The scale is currently called “Attitudes towards Robots Questionnaire” which is a very generic name. However, the scale is very short not covering the entire breadth of the topic of robotics, as one would expect from a generic scale, but seems to be primarily focused on social robots. I think this should be reflected in the title as well, since it leaves out large parts of robotics, such as industrial robotics. I advise the authors to find a different name for their scale.

2.      2. Introduction and Theoretical Background: The introduction and theoretical background in the first part of the manuscript is quite short and does not go deep enough in my opinion. After the increasing research efforts in the field of human-robot interaction in the past few years, there already exist many studies on the topic of attitudes towards robots, not only generic scales like the NARS or the study by Bröhl, Nelles, Brandl, Mertens, & Nitsch (2019), but also specific ones for subareas of robotics or specific robot models like the scale for measuring attitudes towards mobile manufacturing robots (see Leichtmann, Hartung, Wilhelm, & Nitsch, 2023), or the scale for measuring attitudes towards robots in an educational context (Sisman, Gunay, & Kucuk, 2019). I think the current manuscript should more clearly define what the purpose of this new scale is and what it is supposed to predict. Is it supposed to be a generic scale? Is it supposed to be a specific scale for the healthcare context? Is it supposed to measure attitudes toward a specific robot type? And more importantly: How does this new scale or new work fit into the field of existing literature on attitudes towards robots? To which scales is it similar (e.g. NARS), and from which is it different and how (e.g. ACIR-Q)? I think the authors should give a more comprehensive overview on attitudes toward robots here, cite the other work on attitudes toward robots (Bröhl et al., 2019; Leichtmann et al., 2023; Sisman et al., 2019; and others…), and elaborate the similarities and differences, especially concerning all the other scales on attitudes toward robots that exist out there. A more systematic literature review is urgently needed.

3.      3. Scale Development: I think the scale development is a bit vague and the rationale for the item selection is somewhat lacking. The items mentioned here seem a bit eclectic and arbitrary. Normally, scale development is derived theoretically, either because empirical knowledge already exists from the literature or previous qualitative or analytic work, or on the basis of a theoretical model from which items are then systematically derived to ensure that a construct is adequately captured in its breadth and depth. For example, the ACIR-Q developed by Leichtmann et al. (2023) is based on two approaches: They developed items based on 1) a review of other existing scales on attitudes toward manufacturing robots, 2) new aspects found in a qualitative work system analysis to identify aspects of work that could be affected by the robot. Furthermore, it was precisely defined for which kind of robot and for which situations the scale is supposed to be valid. I think the authors should reflect for which situations their scale could be used, what behavior it is supposed to predict, and for which kind of robots they developed their scale. I also advise to scan the literature more thoroughly and identify scales that are closely related to the new scale presented here and scales that clearly differ, so one can clearly understand what it is supposed to measure and what it is not supposed to measure or can be used for. Furthermore, I highly recommend to conduct qualitative studies prior to scale development in order to get a better understanding of people’s worries and hopes coming with robots as done by Meissner, Trübswetter, Conti-Kufner, & Schmidtler (2020), for example. The development of items would then be the next step where items then cover a large part of the identified important beliefs about robots. Thus I think that more studies need be done for validation...

4.      4. Scale Content: I think the content of the current scale is quite confusing. I am not sure what the scale should measure. At the beginning I thought that it is a scale specifically developed for the healthcare context which would be interesting. However, I then realized that only about 3 to 4 items are directed toward this healthcare context. I would have expected items that cover maybe different aspects of healthcare since healthcare in general covers a wide range of tasks – maybe people are willing to allow a robot to do some of these tasks (e.g., reminding to take the medicine) but not others (e.g., carrying a patient) and people vary in the amount of tasks the are comfortable with a robot doing them. However, the content of the other items deal with completely different things such as having sexual contact with a robot implying that these items are directed toward a sexbot which is a completely different type of robot. Another item deals with the concept of “consciousness” of robots. I have a particular problem with the content of this item as 1) I think “consciousness” is a complex construct especially within the domain of robotics and not easy to grasp (however, items should be simple for every person to understand them), 2) it is even unclear within the robotics community whether robots even CAN have some degree of “consciousness” – so the item might not make sense from a professional standpoint, and 3) I am not sure what this item should predict? I have a similar problem with the item on marrying a robot as I do not think that this is very relevant in today’s society – what is this item supposed to predict from a practical standpoint? Why is it important to ask if people are willing to marry a robot? I am not sure if this makes much sense.

5.      5. Empirical Study: Another very strong criticism concerns the empirical study. I think that this study is not sufficient for a validation process and more data and additional studies are urgently needed. Typically, validation processes consist of several sub-studies. For example, while the first study investigates the factor structure exploratively, at least a second study is needed to confirm this structure again via confirmatory factor analysis (with corresponding fit indices such as CFI, TLI, RMSEA, etc.). Furthermore, I think that the amount of data here is too small for this number of tests. Much larger samples would be needed here (I suggest N > 400). I understand that in some cases it can be difficult to achieve large samples if you want to study very specific populations, such as nurses. In such cases of difficult target populations, samples of just over 200 would be justifiable. However, here only an online convenience sample was used. Nowadays, online convenience samples are very easy to test and at low cost. I therefore think that significantly larger samples would be easily possible here. I recommend that the authors conduct another study with a much larger sample before publication. I also recommend that the authors use attention check items to ensure a certain data quality as online samples are especially vulnerable to inattentional responding (see Maniaci, & Rogge, 2014).

6.      6. Validation process: As I mentioned before, I do not think that the work presented here is enough for a validation process. I think more studies are needed to test whether the scale predicts some sort of behavior? Or to test a re-test reliability? How about using different samples such as nurses or how about a sample of people who are actually currently in a nursing home and for whom such technology would be directly relevant? It would be interesting to see if the scale would show differences between certain samples. One could then test for measurement invariance...

7.      7. Open Science: Open Science is a growing trend. I thus encourage the authors to share their data and analysis code, so others can re-run the analysis and test its reproducibility.

I am very sorry that I cannot evaluate the manuscript better and have to recommend a rejection. I think that the work has too many weaknesses and it is too early for publication. I think that the scale is far from validated and it needs a longer process before you can talk about validity. I think that the present study can only represent a first preliminary step in such a process. I would recommend the authors 1) to work out the content of the scale better and more targeted (with the help of qualitative preliminary studies, such as interviews with patients and nurses, or by deriving it from the literature and a theoretical model), 2) to conduct at least two additional studies for the finished scale, one exploratory and then again confirmatory, 3) to test significantly larger samples including data quality tests (I think especially with online convenience samples large samples N >350, are easily possible here), 4) base the introduction and theoretical background less on science fiction robots, but rather on actually used real robots (e.g., NAO, Pepper, ect. – these are already widely used in many empirical studies on human-robot interaction) and give a significantly deeper insight into existing scales on attitudes toward robots currently described in the literature and cite them (e.g., Bröhl et al., 2019; Sisman et al., 2019; Leichtmann et al., 2023; ect.). I think the development of scales is very important, however, I think that this requires more effort. I wish the authors good luck with the next steps and I hope that my recommendations are helpful in this respect. Below I recommend some literature that I think should be cited.

Literature recommendations:

Franke, T., Attig, C., & Wessel, D. (2019). A personal resource for technology interaction: development and validation of the affinity for technology interaction (ATI) scale. International Journal of Human–Computer Interaction, 35(6), 456-467. https://doi.org/10.1080/10447318.2018.1456150

Bröhl, C., Nelles, J., Brandl, C., Mertens, A., & Nitsch, V. (2019). Human–robot collaboration acceptance model: development and comparison for Germany, Japan, China and the USA. International Journal of Social Robotics, 11(5), 709-726. https://doi.org/10.1007/s12369-019-00593-0

Sisman, B., Gunay, D., & Kucuk, S. (2019). Development and validation of an educational robot attitude scale (ERAS) for secondary school students. Interactive Learning Environments, 27(3), 377-388. https://doi.org/10.1080/10494820.2018.1474234

Andtfolk, M., Nyholm, L., Eide, H., Rauhala, A., & Fagerström, L. (2021). Attitudes toward the use of humanoid robots in healthcare—a cross-sectional study. AI & SOCIETY, 1-10. https://doi.org/10.1007/s00146-021-01271-4

Leichtmann, B., Hartung, J., Wilhelm, O., & Nitsch, V. (2023). New short scale to measure workers’ attitudes toward the implementation of cooperative robots in industrial work settings: Instrument development and exploration of attitude structure. International Journal of Social Robotics, 1-22. https://doi.org/10.1007/s12369-023-00996-0

Meissner, A., Trübswetter, A., Conti-Kufner, A. S., & Schmidtler, J. (2020). Friend or foe? understanding assembly workers’ acceptance of human-robot collaboration. ACM Transactions on Human-Robot Interaction (THRI), 10(1), 1-30. https://doi.org/10.1145/3399433

Maniaci, M. R., & Rogge, R. D. (2014). Caring about carelessness: Participant inattention and its effects on research. Journal of Research in Personality, 48, 61-83.

Author Response

The manuscript describes the development of a new scale to measure attitudes toward social robots predominantly in the context of healthcare and the results of a study with 217 participants reporting factor analysis and correlation with other constructs. In principle, of course, the attempt to develop a scale is worthwhile, but the procedure has many major weaknesses in the development and the attempt to validate it. In general, I think that the scale was not properly constructed as it was neither guided by theoretical models nor on the basis of qualitative empirical experience, and also the validation process is based on too little data (only one study with an online convenience sample of only 217 participants seems very small). I am thus not convinced that the scale is reliable or valid. I think, therefore, a lot of work is still necessary before publication including the collection of further data in additional studies. I think that the necessary improvements I propose cannot be made in a reasonable period of time. I thus unfortunately recommend a rejection. In the following I give tips for improvement, which I hope will help the authors in their future scale construction and validation.

  1. 1. Title: First, I think the title and the name of the scale are misleading. The scale is currently called “Attitudes towards Robots Questionnaire” which is a very generic name. However, the scale is very short not covering the entire breadth of the topic of robotics, as one would expect from a generic scale, but seems to be primarily focused on social robots. I think this should be reflected in the title as well, since it leaves out large parts of robotics, such as industrial robotics. I advise the authors to find a different name for their scale.

Authors comment: Indeed it is about social robots. Thus we now call it “Attitudes towards Social Robots Scale” (ASRS). We also explain our approach in more detail below.

  1. Introduction and Theoretical Background: The introduction and theoretical background in the first part of the manuscript is quite short and does not go deep enough in my opinion. After the increasing research efforts in the field of human-robot interaction in the past few years, there already exist many studies on the topic of attitudes towards robots, not only generic scales like the NARS or the study by Bröhl, Nelles, Brandl, Mertens, & Nitsch (2019), but also specific ones for subareas of robotics or specific robot models like the scale for measuring attitudes towards mobile manufacturing robots (see Leichtmann, Hartung, Wilhelm, & Nitsch, 2023), or the scale for measuring attitudes towards robots in an educational context (Sisman, Gunay, & Kucuk, 2019). I think the current manuscript should more clearly define what the purpose of this new scale is and what it is supposed to predict. Is it supposed to be a generic scale? Is it supposed to be a specific scale for the healthcare context? Is it supposed to measure attitudes toward a specific robot type? And more importantly: How does this new scale or new work fit into the field of existing literature on attitudes towards robots? To which scales is it similar (e.g. NARS), and from which is it different and how (e.g. ACIR-Q)? I think the authors should give a more comprehensive overview on attitudes toward robots here, cite the other work on attitudes toward robots (Bröhl et al., 2019; Leichtmann et al., 2023; Sisman et al., 2019; and others…), and elaborate the similarities and differences, especially concerning all the other scales on attitudes toward robots that exist out there. A more systematic literature review is urgently needed.

Authors comment: We now have added the respective literature, give a short overview on the respective scales (a bigger one is provided by Naneva et al. 2020), which we now cite and refer to. We also contextualize our scale into this context. 

  1. Scale Development: I think the scale development is a bit vague and the rationale for the item selection is somewhat lacking. The items mentioned here seem a bit eclectic and arbitrary. Normally, scale development is derived theoretically, either because empirical knowledge already exists from the literature or previous qualitative or analytic work, or on the basis of a theoretical model from which items are then systematically derived to ensure that a construct is adequately captured in its breadth and depth. For example, the ACIR-Q developed by Leichtmann et al. (2023) is based on two approaches: They developed items based on 1) a review of other existing scales on attitudes toward manufacturing robots, 2) new aspects found in a qualitative work system analysis to identify aspects of work that could be affected by the robot. Furthermore, it was precisely defined for which kind of robot and for which situations the scale is supposed to be valid. I think the authors should reflect for which situations their scale could be used, what behavior it is supposed to predict, and for which kind of robots they developed their scale. I also advise to scan the literature more thoroughly and identify scales that are closely related to the new scale presented here and scales that clearly differ, so one can clearly understand what it is supposed to measure and what it is not supposed to measure or can be used for. Furthermore, I highly recommend to conduct qualitative studies prior to scale development in order to get a better understanding of people’s worries and hopes coming with robots as done by Meissner, Trübswetter, Conti-Kufner, & Schmidtler (2020), for example. The development of items would then be the next step where items then cover a large part of the identified important beliefs about robots. Thus I think that more studies need be done for validation...

Authors comment: Indeed the item selection was poorly described and we now elaborate on this aspect more comprehensively. We also describe for which kind of robot (à social robots) and for which situations (Everyday life/coexistence) the scale is supposed to be valid. We also cited the work, which was relevant for our approach (Levy, 2016). Unfortunately we did not rely on qualitative studies in advance. This however does not imply that the scale is not valid (see i.e. the ERAS scale of Sisman et al.) which to our knowledge also used a quite straightforward approach. Although our answer might not be satisfying, we still hope that our psychometric results may be convincing.

  1. Scale Content: I think the content of the current scale is quite confusing. I am not sure what the scale should measure. At the beginning I thought that it is a scale specifically developed for the healthcare context which would be interesting. However, I then realized that only about 3 to 4 items are directed toward this healthcare context. I would have expected items that cover maybe different aspects of healthcare since healthcare in general covers a wide range of tasks – maybe people are willing to allow a robot to do some of these tasks (e.g., reminding to take the medicine) but not others (e.g., carrying a patient) and people vary in the amount of tasks the are comfortable with a robot doing them. However, the content of the other items deal with completely different things such as having sexual contact with a robot implying that these items are directed toward a sexbot which is a completely different type of robot. Another item deals with the concept of “consciousness” of robots. I have a particular problem with the content of this item as 1) I think “consciousness” is a complex construct especially within the domain of robotics and not easy to grasp (however, items should be simple for every person to understand them), 2) it is even unclear within the robotics community whether robots even CAN have some degree of “consciousness” – so the item might not make sense from a professional standpoint, and 3) I am not sure what this item should predict? I have a similar problem with the item on marrying a robot as I do not think that this is very relevant in today’s society – what is this item supposed to predict from a practical standpoint? Why is it important to ask if people are willing to marry a robot? I am not sure if this makes much sense.

Authors comment: Indeed the scale was not intended to measure specific aspects of Healthcare. However, we strongly believe that everyday life has a strong connection to public health and the health care sector. As mentioned above (and now in the manuscript, our idea was to cover the aspects of a social robot as companion and/or carer/helper as described in  (Levy, 2016)

With respect to the complexity of the topic, we do agree. However other scales in the field of Human Robot Interactions also operate with items like “Most robots make poor teammates” (Schaefer, 2020) or “Robots can be trusted” (Koverola, 2020). In particular, the last example also operates with the complex construct of “trust” which also implies an idea of consciousness behind. And indeed, marrying or having sex, which together with the other “caring” items in our idea does not have a tendency towards a sexbot (see e.g. Walter 2019) for a legal and philosophical assessment of this idea. Our interpretation points towards the idea of have a closer partnership with a robot.  Finally our results show a high consistency of the items which should be taken into account. Thus we strongly believe  that our approach and  our interpretation does not point into a direction which “does not make sense”. However, we are grateful for your points as they should be taken into account when discussing our finding. We now have included them into our paper in the discussion part.  

  1. Empirical Study: Another very strong criticism concerns the empirical study. I think that this study is not sufficient for a validation process and more data and additional studies are urgently needed. Typically, validation processes consist of several sub-studies. For example, while the first study investigates the factor structure exploratively, at least a second study is needed to confirm this structure again via confirmatory factor analysis (with corresponding fit indices such as CFI, TLI, RMSEA, etc.). Furthermore, I think that the amount of data here is too small for this number of tests. Much larger samples would be needed here (I suggest N > 400). I understand that in some cases it can be difficult to achieve large samples if you want to study very specific populations, such as nurses. In such cases of difficult target populations, samples of just over 200 would be justifiable. However, here only an online convenience sample was used. Nowadays, online convenience samples are very easy to test and at low cost. I therefore think that significantly larger samples would be easily possible here. I recommend that the authors conduct another study with a much larger sample before publication. I also recommend that the authors use attention check items to ensure a certain data quality as online samples are especially vulnerable to inattentional responding (see Maniaci, & Rogge, 2014).

Authors comment: Thank you for your methodological comment. Indeed we did not run a outlined prestudy to develop the items. However, we still believe that our scale is reliable and valid, which is underpinned by the empirical results, i.e. Cronbachs’s alpha McDonalds Omega or explained variance. We however have taken your comment as a very serious concern and thus have included these aspects in our discussion. Another aspect deals with the sample size. Our study in its size is comparable to other scale developmental studies e.g. the ERAS (Sisman et al. 2019), in which the sample of the study comprised of 232 secondary school students. Methodologically according to Guadagnoli and Velicer (1988) as well as Osborne and Costello (2009) a number of 10 participants per item is recommended for factor analytical approaches. Thus, with 214 participants (3 were excluded due to age lower 18 years), the sample size is absolutely sufficient for a 8 item scale. It moreover also is sufficient for univariate statistics (i.e correlations and ANOVA). We are grateful for pointing on these aspects and have added them into the methods section.

  1. Validation process: As I mentioned before, I do not think that the work presented here is enough for a validation process. I think more studies are needed to test whether the scale predicts some sort of behavior? Or to test a re-test reliability? How about using different samples such as nurses or how about a sample of people who are actually currently in a nursing home and for whom such technology would be directly relevant? It would be interesting to see if the scale would show differences between certain samples. One could then test for measurement invariance.

Authors comment: Thank you again for your comment on the validity of the scale. Indeed studies like you recommended in other samples and to test measurement invariance should be done. For the present special issue we will be unable to run those validation studies. We nevertheless have included them in our conclusion section.  

  1. Open Science: Open Science is a growing trend. I thus encourage the authors to share their data and analysis code, so others can re-run the analysis and test its reproducibility.

Authors comment: Thank you for your suggestion. Open Science is an important issue. Due to the nature of this study at the time of conduction, data can be obtained for reanalysis on request.

I am very sorry that I cannot evaluate the manuscript better and have to recommend a rejection. I think that the work has too many weaknesses and it is too early for publication. I think that the scale is far from validated and it needs a longer process before you can talk about validity. I think that the present study can only represent a first preliminary step in such a process. I would recommend the authors 1) to work out the content of the scale better and more targeted (with the help of qualitative preliminary studies, such as interviews with patients and nurses, or by deriving it from the literature and a theoretical model), 2) to conduct at least two additional studies for the finished scale, one exploratory and then again confirmatory, 3) to test significantly larger samples including data quality tests (I think especially with online convenience samples large samples N >350, are easily possible here), 4) base the introduction and theoretical background less on science fiction robots, but rather on actually used real robots (e.g., NAO, Pepper, ect. – these are already widely used in many empirical studies on human-robot interaction) and give a significantly deeper insight into existing scales on attitudes toward robots currently described in the literature and cite them (e.g., Bröhl et al., 2019; Sisman et al., 2019; Leichtmann et al., 2023; ect.). I think the development of scales is very important, however, I think that this requires more effort. I wish the authors good luck with the next steps and I hope that my recommendations are helpful in this respect. Below I recommend some literature that I think should be cited.

Authors comment: We have tried our best to give an extensive rebuttal of your points, which indeed cannot be handled in this study and this manuscript alone. As we were given a chance to revise our manuscript, we have included your points and your literature and hope this might reflect our attitude to honestly revise the manuscript to the best of our capacities.

Reviewer 2 Report

Comments and Suggestions for Authors

General comment: the article presents itself as an interesting survey aimed at developing a scale to measure attitudes towards robots. This survey is particularly useful as robotic solutions have become increasingly popular in recent years and it is of paramount importance to determine what people's attitudes towards them are, so one is always looking for tools that can help in this regard. I found the part of the introduction with too much emphasis on the humanistic-philosophical part of the question (especially the initial part on the review of robots and the final part, dense with concepts of psychology). They seem to miss the central issue of the article, namely the need to construct a questionnaire to measure people's attitudes towards robots. This part should, in my opinion, be revised, reducing the initial and final information in favour of a framework more aimed at making the reader understand the authors' intention, which in my opinion is somewhat lost. In general, the discussion part appears to be the weakest part of the article: what was reported in the results part should be reported in a more narrative form, explaining it. For a better understanding, it might be useful to include an initial part describing what emerged, then linking it to other research/studies, otherwise the risk is that one loses the thread of the discourse and does not understand the results. I would suggest some corrections, which might clarify some passages, so as to make the article smoother and more solid.

1.Introduction

Page 2, line 47-49: The authors could expand on this concept, proposing hypotheses and explanations

Page 2, line 60-65: This concept, although clear and well explained, does not seem to be linked to the context, nor to the concepts expressed immediately after. We therefore ask the authors to reformulate this part, trying to better link this concept of developmental psychology, which in itself could be interesting, to what is expressed afterwards

Page 2, line 85: What do you mean by 'more positive' approach?

2.1. Data collection

Page 2, line 89: First write a list of what was administered to get a clear overview

Page 2, line 91: Criteria for inclusion of participants? Or could anyone participate?

Page 2, line 93: How? Was an information meeting conducted, for example?

Page 2-3, line 95-96: Please explain this passage better: did some participants fill in the questionnaires online and others paper and pencil or were some questionnaires online and others paper and pencil?

2.2.1. Attitudes towards Robots Questionnaire (AttRoQ)

Page 3, line 100: How were these items generated?

Page 3, line 101: What kind of experts?

Page 3, line 101: What were the response modes? Multiple choice? Likert scale? Yes/No?

4.Discussion

Page 8, line 260: In my opinion, the actual part in which the results of the study are discussed in a more discursive manner is missing, so that all the subsequent parts of the paragraph also seem disconnected from each other. So I would insert a more descriptive part of the results, so that the reader can understand the results, perhaps supported by some graphs

Page 9, line 291: which participants are we talking about? This paragraph is not very clear, are we talking about the German survey [ref. 44]? How is it related to the results of the article?

Page 9, line 294: Again, it is not clear how the inclusion of this study relates to the research of the present study. The authors are requested to contextualise it

Page 9, line 308: Which personality traits specifically?

5.Limitations

Page 10, line 316: the data show that the average age of the sample was rather low and schooling was high: the authors might reflect on the generalisability of the results and whether the general population was adequately represented or not

Author Response

General comment: the article presents itself as an interesting survey aimed at developing a scale to measure attitudes towards robots. This survey is particularly useful as robotic solutions have become increasingly popular in recent years and it is of paramount importance to determine what people's attitudes towards them are, so one is always looking for tools that can help in this regard. I found the part of the introduction with too much emphasis on the humanistic-philosophical part of the question (especially the initial part on the review of robots and the final part, dense with concepts of psychology). They seem to miss the central issue of the article, namely the need to construct a questionnaire to measure people's attitudes towards robots. This part should, in my opinion, be revised, reducing the initial and final information in favour of a framework more aimed at making the reader understand the authors' intention, which in my opinion is somewhat lost. In general, the discussion part appears to be the weakest part of the article: what was reported in the results part should be reported in a more narrative form, explaining it. For a better understanding, it might be useful to include an initial part describing what emerged, then linking it to other research/studies, otherwise the risk is that one loses the thread of the discourse and does not understand the results. I would suggest some corrections, which might clarify some passages, so as to make the article smoother and more solid.

Authors comment: Thank you very much for this helpful omments. Indeed we have recognized an imbalance in the introductory part and have tried to cover the points raised above in more detail. We also shifted some points from the discussion into the introduction and tried to be more narrative in the discussion part. For any details see the comments below.

1.Introduction

Page 2, line 47-49: The authors could expand on this concept, proposing hypotheses and explanations

Authors comment: Thank you for this suggestion. We now have expanded on this idea.

Page 2, line 60-65: This concept, although clear and well explained, does not seem to be linked to the context, nor to the concepts expressed immediately after. We therefore ask the authors to reformulate this part, trying to better link this concept of developmental psychology, which in itself could be interesting, to what is expressed afterwards

Authors comment: We now have skipped the first sentence and tried to reformulate to have a more visible line of argumentation.

Page 2, line 85: What do you mean by 'more positive' approach?

Authors comment: We now have elaborated on this aspect and also refered to important literature to make our point more clear.

2.1. Data collection

Page 2, line 89: First write a list of what was administered to get a clear overview

Authors comment: We now start with the instruments and have put the data collection behind.

Page 2, line 91: Criteria for inclusion of participants? Or could anyone participate?

Authors comment: We now have added the inclusion criteria.

Page 2, line 93: How? Was an information meeting conducted, for example?

Authors comment: An information meeting was not conducted

Page 2-3, line 95-96: Please explain this passage better: did some participants fill in the questionnaires online and others paper and pencil or were some questionnaires online and others paper and pencil?

Authors comment: We now have explained it better: some participants fill in the questionnaires online and others paper and pencil

2.2.1. Attitudes towards Robots Questionnaire (AttRoQ)

Page 3, line 100: How were these items generated?

Authors comment: We now have explained how the items were generated.

Page 3, line 101: What kind of experts?

Authors comment: The expert panel consisted of health care experts, psychologists and computer scientists. We have added this information.

Page 3, line 101: What were the response modes? Multiple choice? Likert scale? Yes/No?

Authors comment: For all items a 6 point Likert scale was used ranging from “1=Absolutely disagree to 6 = absolutely agree. We have added this information.

4.Discussion

Page 8, line 260: In my opinion, the actual part in which the results of the study are discussed in a more discursive manner is missing, so that all the subsequent parts of the paragraph also seem disconnected from each other. So I would insert a more descriptive part of the results, so that the reader can understand the results, perhaps supported by some graphs

Authors comment: According to the journal style, graphical elements in the discussion part are very uncommon. However, we inserted a descriptive part of the results.

Page 9, line 291: which participants are we talking about? This paragraph is not very clear, are we talking about the German survey [ref. 44]? How is it related to the results of the article?

Authors comment: Indeed this part is quite unclear and we have reformulated it.

Page 9, line 294: Again, it is not clear how the inclusion of this study relates to the research of the present study. The authors are requested to contextualise it

Authors comment: We tried our best to contextualize

Page 9, line 308: Which personality traits specifically?

Authors comments: The personality traits are directly listed afterwards. Thus we believe the sentence can remain as it is.

5.Limitations

Page 10, line 316: the data show that the average age of the sample was rather low and schooling was high: the authors might reflect on the generalisability of the results and whether the general population was adequately represented or not

Authors comment: Indeed this is true and thus, our factorial solution might not hold for a general population or for other subsamples like elderly people. We have added this as a limitation and recommend further studies in the conclusion.

Reviewer 3 Report

Comments and Suggestions for Authors

This was an interesting study and manuscript, which had scope for further enhancements.

1. The data seem to be from 2018 (five years ago). Because of the fast-moving world of robotics, these data could be out of date. This would at least need to be discussed as a major limitation, and could even make the manuscript of reduced value. It also creates some potential asynchrony between the statements in the text and the situation on the ground at the time of data collection (e.g. the text claims there is widespread use of robot mowers and floor cleaners - were these widespread in 2018?).

2. There are ethical issues surrounding the lack of approval by an ethics committee of this study involving human participants, particularly with a questions about sex with robots being answered by 15-year-old minors. This may be a very serious issue, which I would recommend the journal looks into. It seems important to align with international norms on this important dimension of the research.

3. In the method, please specify for all scales (particularly your own) what the response options were and how these map onto the numerical scores. Without that information the data cannot be interpreted.

4. External validity: Relationship with other constructs - This header suggests that the other measures (other than the robot attitudes scale) provide validation, but that is not the case for the measures reported here. The  correlates (e.g. personality, optimism) are not really measures by which the robot attitude scale can be validated, but additional measures of separate scientific interest. It would be important to correct that in the text. It leaves the scale somewhat under-validated, because there are not data that allow the researchers to check for convergent validity with a previously developed scale that measures a similar construct. This should be discussed as a limitation.

5. Adapt any numbers reported so that . is used to indicate decimals instead of ,   (to match the English convention, rather than the European).

6. Tables should have brief titles and explanatory notes should be underneath (at least in most journal styles; ignore if not applicable here). The note from Table 4 seems to have got separated from the table.

7. Item difficulty does not make sense as a measure in an attitude scale. It is more relevant for instruments that measure ability. I would suggest dropping all reference / reporting of this measure from the text and tables.

8. Please provide more detail about the exclusion criteria (did even a single missing data point merit exclusion or was it a certain threshold percentage of missing responses for the excluded participants?). 113/330 excluded seems like a very high exclusion rate. Was there scope to estimate missing data for some of these exclusions? Provide enough detail to permit replication.

9. It is rare to validate a scale from scratch and not to have to drop items that do not load onto either factor or cross-load onto more than one factor. Was there any prior piloting, or was there some other work that made this happen? It could be best to report this for transparency and replicability.

10. line 207: "Factor loadings of the items ranged between 0.788 and 0.928 without side 207 loadings >0.3 for the second factor."  "Side loadings" > Cross loadings is a more commonly used term.

11. Line 262: "Our factor analytic tests yielded a stable and convergent two-factor solution that exhibited convincing validity with values of  Cronbach's alpha of 0.9." > The alpha for the second subscale was lower than that. Please correct this, as it is misleading to state this here.

12. There seems to be a more recent scale measuring a very similar construct [reference 47] that is not acknowledged until the Discussion, but this should have been discussed in the Introduction (maybe the long time gap between the data collection / design / background research evaluation and the production of the manuscript created this situation, but it is best to write it from the current time perspective). 

13. The use of "also" in some of the items may bias to repeating the same response. Please discuss this in the limitations. Some additional analysis may be useful (e.g. do items that have "also" in their phrasing correlate more strongly with the previous item than with all other items from the same subscale?).

14. It could be useful to include the German phrasing of the scale items in an Appendix or Supplementary Files online for interested readers.

15. There are some researchers who would find PCA a less suitable method of analysis in this context than Exploratory Factor Analysis. Some brief justification of the choice may be useful.

Comments on the Quality of English Language

a) There are a few ungrammaticalities and unusual word choices, and having some help fixing these would enhance the manuscript.

b) A few German words slipped in, I think. I saw "und" and "reacualized" may be a German word (I did not know it and could not find it in  English dictionaries).

Author Response

This was an interesting study and manuscript, which had scope for further enhancements.

  1. The data seem to be from 2018 (five years ago). Because of the fast-moving world of robotics, these data could be out of date. This would at least need to be discussed as a major limitation, and could even make the manuscript of reduced value. It also creates some potential asynchrony between the statements in the text and the situation on the ground at the time of data collection (e.g. the text claims there is widespread use of robot mowers and floor cleaners - were these widespread in 2018?).

Authors comment: Indeed the data is from 2018 and attitudes may have changed. However according to research on measurement invariance over time of scales and constructs e.g. in the field of middle school students’ attitudes toward computer programming (Ober et al., 2023) shows that although scale mean values might change the structure below is stable. We have added this aspect in our discussion.

  1. There are ethical issues surrounding the lack of approval by an ethics committee of this study involving human participants, particularly with a questions about sex with robots being answered by 15-year-old minors. This may be a very serious issue, which I would recommend the journal looks into. It seems important to align with international norms on this important dimension of the research.

Authors comment: Thank you for this important topic. And yes, we had three participants under 18 years, which we falsely included in our analysis. We now excluded them and recalculated the complete analysis. In addition, we also explained that this survey in its initial phase was part of the test construction class and thus, ethical issues have been discussed there.  

  1. In the method, please specify for all scales (particularly your own) what the response options were and how these map onto the numerical scores. Without that information the data cannot be interpreted.

Authors comment: We now have provided the respective information.

  1. External validity: Relationship with other constructs - This header suggests that the other measures (other than the robot attitudes scale) provide validation, but that is not the case for the measures reported here. The correlates (e.g. personality, optimism) are not really measures by which the robot attitude scale can be validated, but additional measures of separate scientific interest. It would be important to correct that in the text. It leaves the scale somewhat under-validated, because there are not data that allow the researchers to check for convergent validity with a previously developed scale that measures a similar construct. This should be discussed as a limitation.

Authors comment: We have rephrased the respective paragraph and added the lack of a sound validation into the limitation section.

  1. Adapt any numbers reported so that . is used to indicate decimals instead of , (to match the English convention, rather than the European).

Authors comment: We have corrected them

  1. Tables should have brief titles and explanatory notes should be underneath (at least in most journal styles; ignore if not applicable here). The note from Table 4 seems to have got separated from the table.

Authors comment: We have corrected table 4.

  1. Item difficulty does not make sense as a measure in an attitude scale. It is more relevant for instruments that measure ability. I would suggest dropping all reference / reporting of this measure from the text and tables.

  Authors comment: We have deleted item difficulty now.

  1. Please provide more detail about the exclusion criteria (did even a single missing data point merit exclusion or was it a certain threshold percentage of missing responses for the excluded participants?). 113/330 excluded seems like a very high exclusion rate. Was there scope to estimate missing data for some of these exclusions? Provide enough detail to permit replication.

Authors comment: Almost all of the participants did not finalize the survey and had missing values in more than 20% of the items. As sample size was sufficient we abstained from analyzing the missing data.

  1. It is rare to validate a scale from scratch and not to have to drop items that do not load onto either factor or cross-load onto more than one factor. Was there any prior piloting, or was there some other work that made this happen? It could be best to report this for transparency and replicability.

Authors comment: Indeed there was one item “Robots are humans of second class”, which we had to drop due to insufficient factor loadings and we now report on this issue.

  1. line 207: "Factor loadings of the items ranged between 0.788 and 0.928 without side 207 loadings >0.3 for the second factor." "Side loadings" > Cross loadings is a more commonly used term.

  Authors comment: We now corrected it.

  1. Line 262: "Our factor analytic tests yielded a stable and convergent two-factor solution that exhibited convincing validity with values of Cronbach's alpha of 0.9." > The alpha for the second subscale was lower than that. Please correct this, as it is misleading to state this here.

  Authors comment: We now corrected it.

  1. There seems to be a more recent scale measuring a very similar construct [reference 47] that is not acknowledged until the Discussion, but this should have been discussed in the Introduction (maybe the long time gap between the data collection / design / background research evaluation and the production of the manuscript created this situation, but it is best to write it from the current time perspective).

Authors comment: Thank you for this important comment. We now have expanded the introduction and also added other scales to get a broader picture of our aim.

  1. The use of "also" in some of the items may bias to repeating the same response. Please discuss this in the limitations. Some additional analysis may be useful (e.g. do items that have "also" in their phrasing correlate more strongly with the previous item than with all other items from the same subscale?).

Authors comment: We rechecked the items and there is only one “also” in it concerning the “marriage” items 8. We have picked up this issue and now report on the correlation with item 7.

  1. It could be useful to include the German phrasing of the scale items in an Appendix or Supplementary Files online for interested readers.

Authors comment: We now have added the German Phrasing of the scale as Appendix.

  1. There are some researchers who would find PCA a less suitable method of analysis in this context than Exploratory Factor Analysis. Some brief justification of the choice may be useful.

Authors comment: Thank you again for this important comment. We have added it into the discussion part.

Comments on the Quality of English Language

  1. a) There are a few ungrammaticalities and unusual word choices, and having some help fixing these would enhance the manuscript.

  1. b) A few German words slipped in, I think. I saw "und" and "reacualized" may be a German word (I did not know it and could not find it in English dictionaries).

Authors comment: We now went through the manuscript again and rephrased if necessary.